# The Trochlear Groove of a Femoral Component Designed for Kinematic Alignment Is Lateral to the Quadriceps Line of Force and Better Laterally Covers the Anterior Femoral Resection Than a Mechanical Alignment Design

**DOI:** 10.3390/jpm12101724

**Published:** 2022-10-16

**Authors:** Elliot Sappey-Marinier, Stephen M. Howell, Alexander J. Nedopil, Maury L. Hull

**Affiliations:** 1Department of Orthopaedic Surgery, University of California, San Francisco, CA 94143, USA; 2Department of Biomedical Engineering, University of California, Davis, CA 95616, USA; 3Orthopädische Klinik König-Ludwig-Haus, Lehrstuhl für Orthopädie der Universität Würzburg, 97074 Würzburg, Germany; 4Department of Orthopedic Surgery, University of California, Davis, CA 95818, USA

**Keywords:** total knee arthroplasty, lateral trochlear undercoverage, prosthetic design, kinematic alignment, patellofemoral relationship

## Abstract

Background: A concern about kinematically aligned (KA) total knee arthroplasty (TKA) is that it relies on femoral components designed for mechanical alignment (MAd-FC) that could affect patellar tracking, in part, because of a trochlear groove orientation that is typically 6° from vertical. KA sets the femoral component coincident to the patient’s pre-arthritic distal and posterior femoral joint lines and restores the Q-angle, which varies widely. Relative to KA and the native knee, aligning the femoral component with MA changes most distal joint lines and Q-angles, and rotates the posterior joint line externally laterally covering the anterior femoral resection. Whether switching from a MAd- to a KAd-FC with a wider trochlear groove orientation of 20.5° from vertical results in radiographic measures known to promote patellar tracking is unknown. The primary aim was to determine whether a KAd-FC sets the trochlear groove lateral to the quadriceps line of force (QLF), better laterally covers the anterior femoral resection, and reduces lateral patella tilt relative to a MAd-FC. The secondary objective was to determine at six weeks whether the KAd-FC resulted in a higher complication rate, less knee extension and flexion, and lower clinical outcomes. Methods: Between April 2019 and July 2022, two surgeons performed sequential bilateral unrestricted caliper-verified KA TKA with manual instruments on thirty-six patients with a KAd- and MAd-FC in opposite knees. An observer measured the angle between a line best-fit to the deepest valley of the trochlea and a line representing the QLF that indicated the patient’s Q-angle. When the trochlear groove was lateral or medial relative to the QLF, the angle is denoted + or −, and the femoral component included or excluded the patient’s Q-angle, respectively. Software measured the lateral undercoverage of the anterior femoral resection on a Computed Tomography (CT) scan, and the patella tilt angle (PTA) on a skyline radiograph. Complications, knee extension and flexion measurements, Oxford Knee Score, KOOS Jr, and Forgotten Joint Score were recorded pre- and post-operatively (at 6 weeks). A paired Student’s T-test determined the difference between the KA TKAs with a KAd-FC and MAd-FC with a significance set at *p* < 0.05. Results: The final analysis included thirty-five patients. The 20.5° trochlear groove of the KAd-FC was lateral to the QLF in 100% (15 ± 3°) of TKAs, which was greater than the 69% (1 ± 3°) lateral to the QLF with the 6° trochlear groove of the MAd-FC (*p* < 0.001). The KAd-FC’s 2 ± 1.9 mm lateral undercoverage of the anterior femoral resection was less than the 4.4 ± 1.5 mm for the MAd-FC (*p* < 0.001). The PTA, complication rate, knee extension and flexion, and clinical outcome measures did not differ between component designs. Conclusions: The KA TKA with a KAd-FC resulted in a trochlear groove lateral to the QLF that included the Q-angle in all patients, and negligible lateral undercoverage of the anterior femoral resection. These newly described radiographic parameters could be helpful when investigating femoral components designed for KA with the intent of promoting patellofemoral kinematics.

## 1. Introduction

A concern about performing total knee arthroplasty (TKA) with kinematic alignment (KA) is that it uses femoral components (FC) designed for mechanical alignment (MAd-FC) that might not be optimal for patellofemoral tracking [1,2,3]. Whereas MA of the femoral component changes most distal joint lines and Q-angles and rotates the posterior joint line externally from the native knee, KA restores the patient’s pre-arthritic distal and posterior femoral joint lines and Q-angle [4,5,6,7]. Although KA does not set the MAd-FC to its intended target, a registry study comparing 7-year follow-up of KA and MA reported little difference in the incidence and type of patellofemoral complications, while some case series reported a small number cases with patella maltracking [8,9]. Hence, there is an interest in evaluating a femoral component explicitly designed for KA (i.e., KAd-FC) with the intent of restoring radiographic characteristics that promote patellofemoral kinematics [1,2,10].

One key feature of any femoral component design is the angle chosen for the trochlear groove, which is a line best-fit to the groove’s deepest valley [7]. For most MAd-FC, the trochlear orientation is set at 6° to approximate the quadriceps line of force (QLF) to promote patellar tracking (Figure 1) [11,12]. 

However, when a surgeon implants a MAd-FC using KA and MA, the 6° trochlear groove could be medial or lateral to the quadriceps line of force, which is directed along the line connecting the anterior inferior iliac spine (AIIS) to the center of the knee [13]. Accordingly, the present study describes a new radiographic measure to assess this difference (Figure 2).

When the trochlear groove is lateral or medial relative to the QLF, the angle is denoted + or −, and the femoral component includes or excludes the patient’s Q-angle, respectively. Therefore, one strategy to promote inclusion of the Q-angle is to design a KAd-FC with a wide lateral trochlear orientation that lies lateral to the QLF across the wide range of femoral and tibial phenotypes (Figure 3) [14].

A second newly described radiographic measurement that could assess the effectiveness of a KAd-FC is the extent of lateral coverage of the anterior femoral resection (Figure 4) [2]. 

A study of a MAd-FC aligned in KA showed greater lateral undercoverage of the anterior femoral resection in those patients with the valgus femoral phenotype because of a medial shift of the proximal extension of the trochlear groove orientation relative to those with varus femoral phenotypes. Therefore, the authors suggested a better lateral coverage between the KAd-FC and femoral trochlea resection could reduce the risk of unfavorable patellar kinematics [2].

A third radiographic measurement that could assess the KAd-FC is its effect on the degree of lateral patella tilt. One study compared the patella tilt angle (PTA) of a MAd-FC implanted with KA and MA. It showed significantly greater lateral patellar tilt in the KA group (12 of 93) relative to the MA group (1 of 93), even though both alignments had comparable outcomes scores [16]. Hence, whether a KAd-FC has less lateral patellar tilt than a MAd-FC is of interest.

Accordingly, the present study analyzed sequential bilateral unrestricted caliper-verified KA TKA performed with manual instruments on thirty-six patients with a KAd- and MAd-FC in opposite knees. The primary aim was to determine whether a KAd-FC sets the trochlear groove lateral to the QLF, better laterally covers the anterior femoral resection, and reduces lateral patella tilt relative to a MAd-FC. The secondary objective was to determine at six weeks whether the KAd-FC resulted in a higher complication rate, less knee extension and flexion, and lower clinical outcomes.

## 2. Materials and Methods

After obtaining approval from an institutional review board (Pro00065392), one author (ESM) performed a retrospective analysis of a prospective database and identified 36 patients treated by two surgeons (SMH *n* = 22; AJN *n* = 14) that met the following criteria. 

The study included all patients that (1) underwent an unrestricted caliper-verified KA with a 6° trochlear groove orientation MAd-FC using a posterior cruciate ligament retaining (CR) insert design (GMK Sphere, Medacta International, www.medacta.com accessed on 20 September 2022), and (2) a subsequent contralateral TKA with the only difference being the use of a KAd-FC with a wider 20.5° trochlear groove (GMK SpheriKA). Patients had to have an anteroposterior and lateral rotationally controlled, non weight-bearing, long-leg CT scanogram and axial CT scan of both knees on the day of discharge, and Laurin skyline views and lateral radiographs of both knees at 6 weeks. Each patient fulfilled the Centers for Medicare & Medicaid Services guidelines for medical necessity for TKA treatment and had (1) Kellgren-Lawrence Grade III to IV osteoarthritis; (2) any severity of clinical varus or valgus deformity); (3) and any severity of flexion contracture. Excluded were patients with prior fractures of the knee treated with open-reduction internal fixation, inflammatory or septic arthritis, and lower extremity neurologic disorders.

The technique for performing unrestricted caliper-verified KA TKA with manual instruments through a mid-vastus approach and intraoperatively recording a series of verification checks has been previously described [17]. Briefly, for the femoral component, the internal-external (I-E) and varus-valgus (V-V) rotations and the anterior-posterior (A-P) and proximal-distal (P-D) positions were set coincident with the native distal and posterior joint lines by adjusting the calipered thicknesses of the distal and posterior femoral resections to within 0 ± 0.5 mm of those of the femoral component condyles after compensating for cartilage wear and kerf of the saw blade. The accuracy of the experienced and less experienced surgeon setting the femoral component to the KA target with manual instruments was comparable or better than reported values for mechanical alignment using robotic instrumentation [18,19]. 

For the tibial component, the knee was balanced by adjusting the P-D position, V-V rotation, and the medial slope of the tibial resection to match the patient’s pre-arthritic tibial joint line [17]. Negligible V-V laxity in maximum extension with the spacer block and trial component sets 97% of tibial components within the left-to-right symmetry of the non-osteoarthritic lower limbs [4]. The resection’s tibial slope angle (TSA) was set parallel to the medial joint line by adjusting the plane of an angel wing inserted in the tibial guide with a reported mean difference of 0.7 ± 3.2° [5]. Best-fitting the anatomic baseplate parallel to the cortical rim of the tibial resection set I-E with a reported mean difference from the A-P plane of the knee of 2° external ± 5° [20].

A single observer (ESM) measured all the following radiographic alignment parameters using free image-analysis software (Horos Imaging Software, https://horosproject.org). The method for measuring the angle between the trochlear groove of the KAd- and MAd-femoral component and the QLF is shown in Figure 2. The lateral undercoverage of the anterior femoral resection was measured on a 3D multi-plane reconstruction of the axial CT scan that projected the femoral component parallel to the lateral lug on the sagittal and coronal views and parallel to both lugs on the axial view as shown in Figure 4 [15]. The lateral undercoverage was the distance, in millimeters, between the lateral edge of the femur and prosthetic trochlea at its sagittal midpoint. The PTA was the angle formed by a line tangent to the anterior border of the femoral condyles and a line tangent to the patella-prosthesis interface on the skyline radiographic view (+ lateral tilt, − medial tilt) [15].

Pre-operatively and six weeks after surgery, a long-arm goniometer measured the limit of active knee extension and flexion. In addition, the patient completed the Knee Society Knee and Function scores, Oxford Knee Score (OKS; 48 best, 0 worst), Knee injury and Osteoarthritis Outcome Score (KOOS JR; 100 best, 0 worst). Finally, they completed the Forgotten Joint Score (FJS; 100 best, 0 worst) at six weeks.

### Statistical Analysis

A power analysis using software (G*Power 3.1.9.6 for Mac OS X 10.7 to 12) determined that 34 patients achieved a power = 0.80, using inputs consisting of a Type I error (alpha) of 0.05, 2 mm difference to detect in lateral undercoverage of the anterior femoral resection and a standard deviation of 1.5 mm.

Two methods of statistical analysis determined the consistency of the measurement for each radiographic parameter. The first method used intraclass correlation coefficient (ICC) analyses to determine inter-observer and intra-observer variability of each radiographic parameter. To quantify inter-observer variability, three observers measured twelve randomly selected TKAs. To quantify intra-observer variability, one observer made five measurements on alternating days on five randomly selected TKAs. A two-factor ANOVA, with observer and patient modelled as random effects, was performed for each radiographic parameter. Intra-observer and inter-observer ICCs were computed using the variance components for observer, patient, and error [21]. The second method determined repeatability (i.e., precision of measurement) which was quantified as the square root of the pooled variance for the single observer.

Dependent variables were reported as either the mean ± standard deviation (SD) or the median and interquartile range (IQR). For all dependent variables, software performed a paired Student’s *T*-test to determine the significance of the difference between the paired KA TKAs with contralateral KAd- and MAd-FC. Significance was set at *p* < 0.05.

## 3. Results

Among the cohort, one patient did not undergo a CT scan because of scanner malfunction. Therefore, the analysis included thirty-five patients. In 34 of 35 patients, the implantation of the KAd-FC was after the MAd-FC at a mean of 10 ± 10 months. The time between obtaining the preoperative extension, flexion, and clinical outcome scores and surgery was 4 ± 3 months for the KAd-FC and 3 ± 3 months for the MAd-FC. The mean age was 67 ± 8 years (51 to 83), of which 23 were females. The mean body-mass-index was 33 ± 7 kg/m^2^ (21 to 47). The pre-operative clinical characteristics are listed in Table 1.

The ICCs for reproducibility and repeatability were either excellent or good for the three postoperative radiographic parameters (Table 2).

The angle between the trochlea groove and QLF was available for thirty-two subjects, as a malrotated limb projection prevented measurement in three. The 20.5° trochlear groove of the KAd-FC was lateral to the QLF and included the Q-angle in 100% (15 ± 3°) of TKAs, which was greater than the MAd-FC with the 6° trochlear groove that was lateral to the QLF in 69% (1 ± 3°) (*p* < 0.001) (Figure 5). 

The KAd-FC’s 2 ± 1.8 mm lateral undercoverage of the anterior femoral resection was 2.4 mm less than the 4.4 ± 1.5 mm for the MAd-FC (*p* < 0.001) (Figure 6). 

The KAd-FC’s mean 4 ± 3.7°of lateral patellar tilt was not different from the 3.8 ± 4.3° of lateral patellar tilt for the MAd-FC (*p* = 0.5296). (Figure 7). 

No TKAs with the KAd- or MAd-FC had a complication at six weeks. Furthermore, the mean values for each post-operative clinical characteristic were not significantly different between the KAd- and MAd-FC (Table 3). 

## 4. Discussion

The present study analyzed sequential bilateral unrestricted caliper-verified KA TKA performed with manual instruments on thirty-five patients with a KAd- and MAd-FC in opposite knees. The most important findings that the orientation of the trochlear groove of the KAd-FC was lateral to the QLF in all patients, thereby including the Q-angle and that the lateral undercoverage of the anterior femoral resection was negligible. In addition, monitoring the patellar tilt, complication rate, knee extension and flexion, and clinical outcome scores of the KAd-FC at six weeks did not detect any adverse or beneficial consequences.

The measurement of the angle between the trochlea groove of the femoral component and the QLF analyzes the effect of different trochlear groove orientations and femoral component designs relative to the patient’s Q-angle. Theoretically, a trochlea groove medial to the QLF could medialize the patella relative to the native position, causing stiffness and anterior knee pain. In contrast, a lateral trochlear groove could promote patellar capture. However, the present study detected no clinical benefit or short-coming from using the KAd-FC, possibly due to the short follow-up, small sample size, and activity level. However, several studies reported there is a risk, albeit low, of patellar complications with KA that increase the risk of revision surgery. One prospective, multicenter study of KA performed with image-derived instrumentation and MAd-FC resulted in a higher than anticipated rate of patellar complications which may be due to the poor rotational accuracy of the instrumentation [22]. A case series of 198 KA TKAs with a MAd-FC reported that three TKAs (1.5%) underwent reoperation for anterior knee pain or patellofemoral instability in the subgroup of patients with the more valgus phenotypes. One explanation for reoperation was that the trochlear groove of the MAd-FC has medial to the patient’s Q-angle. Furthermore, they opined that designing a KAd-FC that restores patellofemoral kinematics with all phenotypes, especially the more valgus ones, is a strategy for reducing reoperation risk [23]. 

Improving the lateral coverage of the anterior femoral resection is another strategy for promoting patellofemoral kinematics. One case series of 52 KA TKAs with a MAd-FC reported a 3 mm mean lateral undercoverage of the anterior femoral resection, which increased as the femoral phenotype became more valgus. The three of 52 KA TKA’s with the most valgus femoral phenotype had 5–6 mm of lateral undercoverage [2]. The present study’s 2 mm and 4.4 mm mean lateral undercoverage of the anterior femoral resection’s for the KAd- and MAd-FC was comparable to the 3 mm mean lateral undercoverage in the case series. Hence, there is a need for further studies to determine any beneficial patellofemoral effects of increasing the coverage of the anterior femoral resection with a KAd-FC, especially in those patients with the most valgus femoral phenotypes. 

The KAd-FC resulted in the same degree of lateral patella tilt as the MAd-FC, which is different and comparable to reports from other studies. For example, one study of TKAs with a MAd-FC aligned in KA and MA showed that even though the lateral patellar tilt was significantly greater in the KA group (12 of 93) relative to the MA group (1 of 93), alignments had comparable outcomes scores [16]. In addition, a case series of 42 KA TKAs performed with a MAd-FC reported that patients with a patella tilt within a range of −3° medial to 11° lateral reported good function, so a patella tilt inside this range in not indicator for patella-stabilizing revision surgery [15]. One reason for the low incidence of reoperation for patellofemoral symptoms is that KA with a MAd-FC restores patellofemoral kinematics and contact pressure distribution closer to the native knee than MA with a MAd-FC [24,25]. Another is that the 3-dimensional native trochlear morphology is more closely restored with KA than MA when assessed with a variety of MAd-FC [1,26,27].

Finally, monitoring the complication rate, knee extension, knee flexion, and clinical outcome scores at just six weeks was to detect any adverse consequences from using the KAd-FC, which were negligible. Unfortunately, the short follow-up and carry-over effect from implanting the KAd-FC after the MAd-FC in 34 of 35 patients prevents any conclusion concerning the effectiveness of the KAd-FC. However, the carry-over effect could be inconsequential as the preoperative to 6-week change was comparable for the KAd-FC vs. MAd-FC in terms of extension of 5° vs. 4°, flexion of −1° vs. −2°, Knee Society Knee Score 40 vs. 44 points, Knee Society Function Score of 9 vs. 8 points, and OKS 10 vs. 14 points (i.e., computed from Table 1 and Table 3). Furthermore, the difference between these pairs of motion and outcome scores was below the minimal clinically important difference.

The present study has several limitations that could affect the generalization of the results. First, the trochlear groove orientation and design of the MAd-FC reported in the present study might not apply to other femoral component designs. However, as presented in Figure 1, most of the MAd-FC from the major companies have a similar 6° trochlear groove orientation relative to vertical as the MAd-FC in the present study. Second, clinical follow-up at 6-weeks was short. However, the aim of the study was to assess immediate potential adverse outcomes. Third, three AP scanograms were not measured because of asymmetric rotational projection of the two limbs. Obtaining optimal images required ongoing quality-control review. Fourth, because the study evaluated an anatomically shaped patella the patellofemoral effects of a domed patella implant and non-resurfaced patella remain unknown. Fifth, the PTA measured at six weeks is a non-dynamic static assessment of the patellofemoral joint at one flexion angle that is not a comprehensive assessment of patella kinematics. In addition, the difference in the time to obtain the skyline radiographic view from surgery being six weeks for the KAd-FC and a mean of 10 months for the MAd-FC could confound the interpretation. Finally, the differences in trochlear groove orientation and lateral coverage of the anterior femoral resection between component designs can not be explained by differences in post-operative coronal and axial alignment between limbs. The accuracy of setting the femoral component in varus-valgus, proximal-distal, internal-external, and anterior-posterior in each limb were within ±0.5 mm due to the highly accurate caliper-verified KA with manual instruments (Table 4) [4,18]. In addition, different alignment philosophies that change the patient’s pre-arthritic joint lines and less accurate instrumentation could result in different orientation and coverage relative to KA.

## 5. Conclusions

The KA TKA with a KAd-FC resulted in a trochlear groove lateral to the QLF that included the Q-angle in all patients, and negligible lateral undercoverage of the anterior femoral resection. These newly described radiographic parameters could be helpful when investigating femoral components designed for KA with the intent of promoting patellofemoral kinematics.

## Figures and Tables

**Figure 1 jpm-12-01724-f001:**
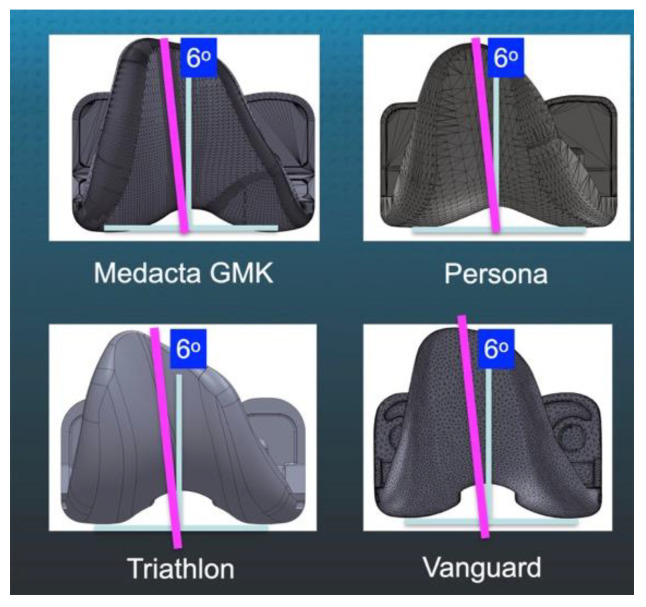
Images of four representative brands of right-sided MAd-FC (femoral component designed for mechanical alignment) show the line (magenta) denoting the trochlear groove that is oriented 6° lateral relative to a line perpendicular (blue) to the distal femoral joint line.

**Figure 2 jpm-12-01724-f002:**
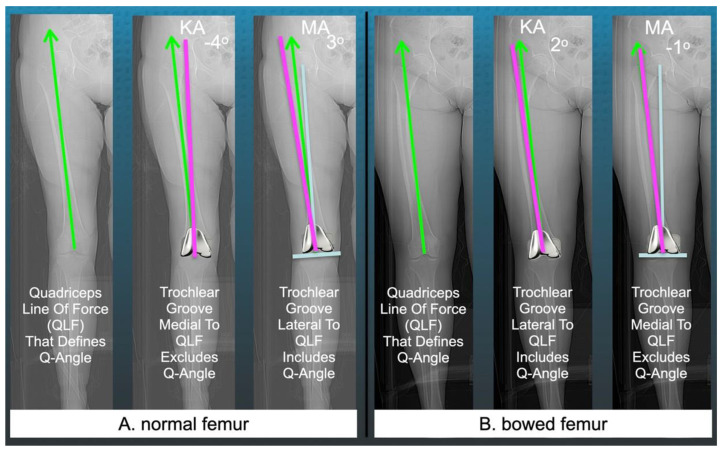
Radiographs of two femurs with a straight diaphysis (**A**) and a lateral bow (**B**) with a femoral component placed with kinematic alignment (KA) and mechanical alignment (MA) showing the new method for measuring the angle between the femoral component’s trochlear groove orientation (magenta line) and the quadriceps line of force (green line) that connects the anterior inferior iliac spine (AIIS) to the center of the knee and comprises the patient’s Q-angle. For MA, the femoral component is placed orthogonal to the femoral mechanical axis (blue line).

**Figure 3 jpm-12-01724-f003:**
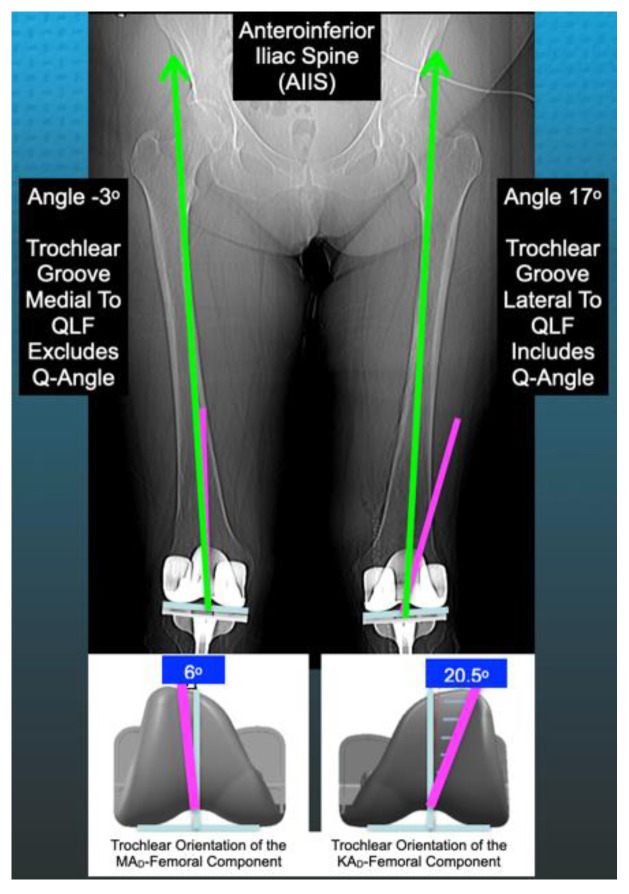
Schematics show the 6° trochlear groove (pink line) of the MAd-FC (**left**) and the 20.5° trochlear groove of the KAd-FC (**right**) evaluated in opposite knees in each patient enrolled in the present study. The angle between the trochlear groove (TG) and the quadriceps line of force (QLF; green line) was measured and defined as TG-QLF. The TG-QLF angle on the scanogram was −3° for the MAd-FC (**left**) and 17° for the KAd-FC (**right**). The distal femoral joint line is defined by the blue line.

**Figure 4 jpm-12-01724-f004:**
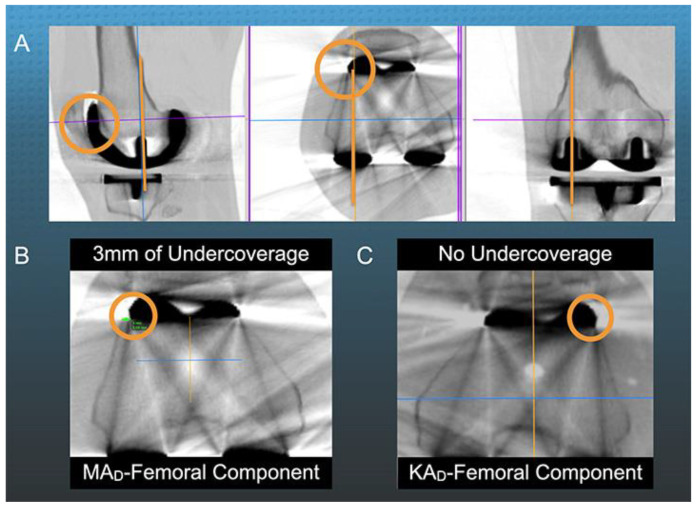
Composite shows the measurement of lateral undercoverage of the anterior femoral resection from a 3D multi-plane reconstruction of an axial computed tomography (CT) scan (**A**) showing 3 mm of undercoverage with KA of the MAd-FC in the right knee (**B**) and no undercoverage with KA of a KAd-FC in the left knee (**C**) [15].

**Figure 5 jpm-12-01724-f005:**
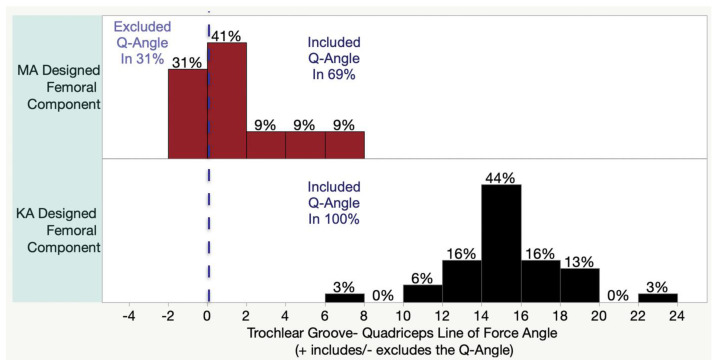
Histograms show the inclusion or exclusion of the patient’s Q-angle for both MAd- and KAd-FC.

**Figure 6 jpm-12-01724-f006:**
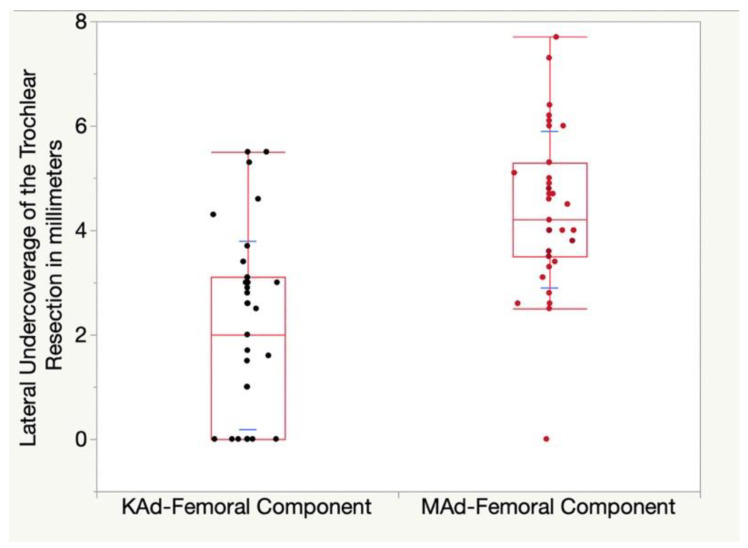
Quantile box plot shows the lateral undercoverage of the anterior femoral resection for the KAd- (black points) and MAd-FC (red points). Standard deviations are illustrated by blue lines.

**Figure 7 jpm-12-01724-f007:**
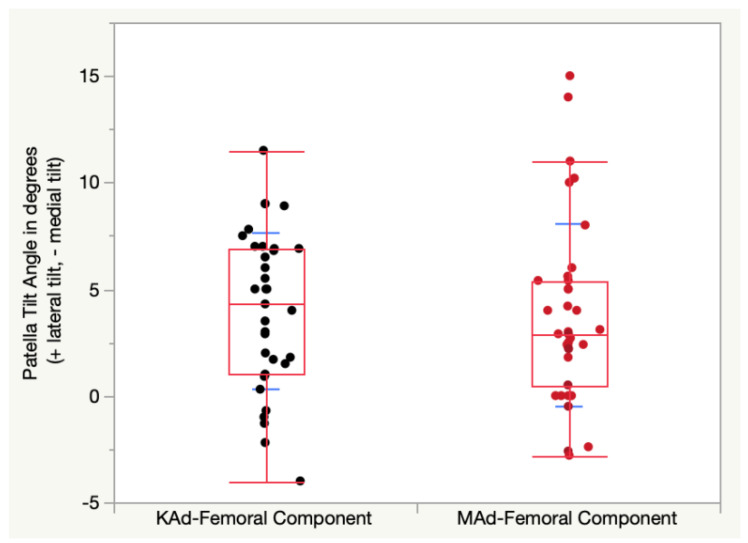
Quantile box plot shows the patellar tilt angle for the KAd- (black points) and MAd-FC (red points). Standard deviations are illustrated by blue lines.

**Table 1 jpm-12-01724-t001:** Pre-operative knee characteristics, and function scores for the knee treated with the KAd- and MAd-FC for the 35 patients analyzed in the present study.

Preoperative Characteristics and Function Scores	KAd-FC Mean ± Standard Deviation (Range)	MAd-FC Mean ± Standard Deviation (Range)	*p*-Value
Extension (°)	8 ± 8° (0 to 36°)	6 ± 6° (0 to 25°)	0.32
Flexion (°)	109 ± 14° (71 to 130°)	108 ± 13° (69 to 128°)	0.86
Type of Osteoarthritic Knee Deformity	80% varus, 5% valgus, 15% patellofemoral	80% varus, 5% valgus, 15% patellofemoral	N.A.
Knee Society Score (100 best, 0 worst)	50 ± 27 points (0 to 100)	41 ± 21 points (0 to 80)	0.14
Knee Society Function Score (100 best, 0 worst)	47 ± 23 points (5 to 90)	45 ± 20 points (0 to 100)	0.62
Oxford Knee Score (48 best, 0 worst)	20 ± 10 points (0 to 36)	18 ± 7 points (5 to 31)	0.34
KOOS Jr (100 best, 0 worst)	44 ± 20 points (0 to 66)	39 ± 14 points (0 to 59)	0.17

**Table 2 jpm-12-01724-t002:** Inter-observer and intra-observer intraclass correlation coefficients (ICC values), repeatability, and agreement classification for measurements of the three postoperative alignment parameters.

Postoperative Alignment Parameter	Inter-Observer Intraclass Correlation	Intra-Observer Intraclass Correlation	Repeatability
Angle Between the Trochlear Groove and Quadriceps Line of Force	ICC = 0.82 ^#^	ICC = 0.81 ^#^	0.2°
Undercoverage of the Lateral Anterior femoral resection	ICC = 0.92 *	ICC = 0.93 *	0.5 mm
Patella Tilt Angle (PTA)	ICC = 0.88 ^#^	ICC = 0.89 ^#^	0.9°

* Excellent Agreement (ICC > 0.9) ^#^ Good Agreement (ICC 0.75 to 0.9).

**Table 3 jpm-12-01724-t003:** Compares range of motion and clinical outcomes scores and complications at 6 weeks post-operative between paired knees treated with KA and a KAd-and MAd-FC.

Clinical Characteristics Assessed Post-Operatively at 6-Weeks	KAd-FC Mean ± SD (Range)	MAd-FC Mean ± SD (Range)	*p*-Value
Extension (°)	2 ± 3° (0 to 10°)	2 ± 3° (0 to 10°)	0.54
Flexion (°)	109 ± 11° (90 to 135°)	107 ± 12° (70 to 126°)	0.11
Knee Society Score Knee (100 best, 0 worst)	91 ± 12 points (40 to 100)	86 ± 12 points (64 to 100)	0.056
Knee Society Score Function (100 best, 0 worst)	57 ± 18 points (0 to 100)	53 ± 18 points (10 to 80)	0.53
Oxford Knee Score (48 best, 0 worst)	30 ± 8 points (8 to 44)	32 ± 8 points (13 to 46)	0.15
KOOS Jr (100 best, 0 worst)	61 ± 14 points (0 to 85)	66 ± 15 points (31 to 100)	0.11
Forgotten Joint Score (100 best, 0 worst)	41 ± 25 points (0 to 94)	52 ± 29 points (0 to 100)	0.08

**Table 4 jpm-12-01724-t004:** Postoperative alignment radiological parameters. All units in degrees.

Radiological Parameters	KAd-FC Mean ± Standard Deviation (Range)	MAd-FC Mean ± Standard Deviation (Range)	*p*-Value
Hip-Knee-Ankle Angle (HKAA)	178 ± 3.2° (170 to 182°)	178 ± 3.3° (171 to 184°)	0.83
Distal Lateral Femoral Angle (DLFA)	88 ± 2.4° (83 to 94°)	88 ± 2.2° (83 to 93°)	0.92
Proximal Medial Tibial Angle (PMTA)	85 ± 2° (82 to 89°)	85 ± 2.6° (79 to 90°)	0.67
Tibial Slope Angle (TSA)	84 ± 2.6° (80 to 89°)	85 ± 3° (75 to 89°)	0.29

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
