# Peer review of "The Trochlear Groove of a Femoral Component Designed for Kinematic Alignment Is Lateral to the Quadriceps Line of Force and Better Laterally Covers the Anterior Femoral Resection Than a Mechanical Alignment Design"

_jpm, 2022, doi:10.3390/jpm12101724_

Round 1

Reviewer 1 Report

Thank you for submitting this interesting manuscript for review. Please see my comments and suggestions below.

General Comment:

If you are using MAD and KAD as abbreviations for the mechanical alignment/kinematic alignment femoral component, there is no need to continue to write ‘femoral component’ after the abbreviation. Please consider correcting this throughout the manuscript.

 Abstract:

Please make the aim and objective of the study clearer, as although it states that 'one question' is addressed, the following line includes a number of variables. The level of significance and statistical analyses carried out on the data should also be described in the abstract.  It is also important to specify the number of datasets that were analysed in the results section. Finally, the conclusion should also be clarified. 

Introduction:

Page 2, Line 62: Please define QLF here.

Please clarify what the primary aim for the study was. A number of variables are listed in the aim, and it is unclear which was the primary aim.

Methods:

How far in the future was the contralateral TKA performed? I am concerned that the results of the second surgery (which appears to have been the KAD procedure in all cases) may have been biased by those of the first. For example, many patient reported outcome measures ask patients about the activities they are able to carry out. If the patient was still recovering from the initial TKA when asked about their second, their perceived ability may be impacted by the initial surgery, even if they are told to think only about the second surgery. Please clarify in the manuscript a) what the average time between surgeries was, b) whether the order of surgery was always the same i.e. MAD > KAD, c) how certain you are that the results from the second surgery are not biased by the outcomes of the first (regardless of time difference between surgeries).

Please also clarify when the pre-operative patient reported outcomes were taken for both knees. Were some of the questionnaires for the second knee taken before/after the first knee was operated on? This is important for the same reason as outlined above.

The methodology should also briefly describe how the ranges of motion were measured for the patients. For example, was the flexion and extension estimated by visually looking at the knee or was it measured with a goniometer? Please also clarify whether the assessment was passive or active.

Discussion:

It would benefit the reader to further understand the implications the limitations of this study may have on the interpretation of the findings. Please expand this section accordingly. 

Please also consider the possibility that the kinematics of the patella may be different at 6-weeks in each component, given the radiological differences. Assessing range of motion should not be a substitute to a biomechanical assessment of the patella's kinematics/kinetics. 

Conclusions:

Please rewrite the conclusions as it is not easy to follow. You should also explain what conclusions were drawn from your study, as opposed to merely stating that more research is necessary. Importantly, what impact might this particular study have?

Tables and Figures:

In the table legends for Table 1 and 2 you can remove ‘Lists shows/lists the’ as these are not necessary. You should also ensure that all legends are descriptions of the content and not a discussion of the findings.

Please explain why there is a p Value for the Complications in Table 3. Surely any complications would have been textual data rather than numeric?

Author Response

Author’s response to reviewer #1

The authors would like to thank the reviewer for his detailed notes and revisions.

Appropriate changes have been made to respond to all reviewer’s revisions. They are highlighted in yellow throughout the manuscript.

We hope that our revisions are received well and that the manuscript remains befitting of publication in your esteemed journal.

Yours sincerely,

The authors.

General Comment:

Comment 1: If you are using MAand KAas abbreviations for the mechanical alignment/kinematic alignment femoral component, there is no need to continue to write ‘femoral component’ after the abbreviation. Please consider correcting this throughout the manuscript.

Authors’ response: As suggested we replaced MAD femoral component and KAD femoral component with MAD-FC and KAD-FC.

Comment 2: Abstract: Please make the aim and objective of the study clearer, as although it states that 'one question' is addressed, the following line includes a number of variables.

Authors’ response: As suggested we changed the aim and objective as follows in the Abstract and Introduction

As follows:

ABSTRACT

The primary aim was to determine whether a KAD-FC sets the trochlear groove lateral to the QLF, better laterally covers the anterior femoral resection, and reduces lateral patella tilt relative to a MAD-FC. The secondary objective was to determine at six weeks whether the KAD-FC resulted in a higher complication rate, less knee extension and flexion, and lower clinical outcomes

INTRODUCTION

The primary aim was to determine whether a KAD-FC sets the trochlear groove lateral to the QLF, better laterally covers the anterior femoral resection, and reduces lateral patella tilt relative to a MAD-FC. The secondary objective was to determine at six weeks whether the KAD-FC resulted in a higher complication rate, less knee extension and flexion, and lower clinical outcomes.

Comment 3: The level of significance and statistical analyses carried out on the data should also be described in the abstract. 

Authors’ response: As suggested we added the type of analysis and significance to the abstract as follows:

A paired Student’s T-test determined the difference between the KA TKAs with a KAD-FC and MAD-FC with a significance set at p<0.05.

Comment 4: It is also important to specify the number of datasets that were analyzed in the results section.

Authors’ response: As suggested we added the number of datasets to the results section as follows:

The final analysis included thirty-five patients.

Comment 5: Finally, the conclusion should also be clarified. 

Authors’ response: As suggested we modified the conclusion as follows:

The KA TKA with a KAD-FC resulted in a trochlear groove lateral to the QLF that included the Q-angle in all patients, and negligible lateral undercoverage of the anterior femoral resection. These newly described radiographic parameters could be helpful when investigating femoral components designed for KA with the intent of promoting patellofemoral kinematics.

Introduction:

Comment 6: Page 2, Line 64: Please define QLF here.

Authors’ response: As suggested we wrote out the keywords for QLF as follows:

Line 64 quadriceps line of force (QLF)

Comment 7: Please clarify what the primary aim for the study was. A number of variables are listed in the aim, and it is unclear which was the primary aim.

Authors’ response: As suggested we changed the aim and objective as follows in the Abstract and Introduction:

INTRODUCTION

The primary aim was to determine whether a KAD-FC sets the trochlear groove lateral to the QLF, better laterally covers the anterior femoral resection, and reduces lateral patella tilt relative to a MAD-FC. The secondary objective was to determine at six weeks whether the KAD-FC resulted in a higher complication rate, less knee extension and flexion, and lower clinical outcomes.

Methods:

Comment 8: How far in the future was the contralateral TKA performed? I am concerned that the results of the second surgery (which appears to have been the KAprocedure in all cases) may have been biased by those of the first. For example, many patient reported outcome measures ask patients about the activities they are able to carry out. If the patient was still recovering from the initial TKA when asked about their second, their perceived ability may be impacted by the initial surgery, even if they are told to think only about the second surgery. In addition clarify whether the order of surgery was always the same i.e. MA> KAD,

Authors’ response: As suggested we added the sequence of the MAD-FC vs KAD-FC, and the mean time  between obtaining the preoperative extension, flexion, and clinical outcome scores to the date of surgery in the Results Section.

RESULTS

In 34 of 35 patients, the implantation of the KAD-FC was after the MAD-FC at a mean of 10 ± 10 months. The time between obtaining the preoperative extension, flexion, and clinical outcome scores and surgery was 4 ± 3 months for the KAD-FC and 3 ± 3 months for the MAD-FC.

Authors’ response: As suggested we added a section in the Discussion that discusses the possibility of a carry-over effect as follows:

DISCUSSION

Finally, monitoring the complication rate, knee extension, knee flexion, and clinical outcome scores at just six weeks was intended to detect any adverse consequences from using the the KAD-FC, which were negligible. Unfortunately, the short follow-up and carry-over effect from implanting the KAD-FC after the MAD-FC in 34 of 35 patients prevents any conclusion concerning the effectiveness of the KAD-FC. However, the carry-over effect could be inconsequential as the preoperative to 6-week change was comparable for the KAD-FC vs. MAD-FC in terms of extension of 5° vs. 4°, flexion of -1° vs. -2°, KSS 40 vs. 44 points, Knee Society Function Score of 9 vs. 8 points, and OKS 10 vs. 14 points (i.e., computed from Table 1 and 3). Furthermore, the difference between these pairs of motion and outcome scores was below the minimal clinically important difference.

Comment 9 : how certain you are that the results from the second surgery are not biased by the outcomes of the first (regardless of time difference between surgeries). 

Authors’ response: Because of the carry-over effect we deleted any reference to the clinical outcome in the conclusion in the Abstract and at the end of the paper, which now reads as follows:

These newly described radiographic parameters could be helpful when investigating femoral components designed for KA with the intent of promoting patellofemoral kinematics.

Comment 10: Please also clarify when the pre-operative patient reported outcomes were taken for both knees. Were some of the questionnaires for the second knee taken before/after the first knee was operated on? This is important for the same reason as outlined above.

Authors’ response: As suggested, we clarified the time between the preoperative evaluation and surgery in the Results section as follows:

The time between obtaining the preoperative extension, flexion, and clinical outcome scores and surgery was 4 ± 3 months for the KAD-FC and 3 ± 3 months for the MAD-FC.

Comment 11: The methodology should also briefly describe how the ranges of motion were measured for the patients. For example, was the flexion and extension estimated by visually looking at the knee or was it measured with a goniometer? Please also clarify whether the assessment was passive or active.

Authors’ response: As suggested, we clarified the method used for measuring range of motion goniometer as follows:

Pre-operatively and six weeks after surgery, a long-arm goniometer measured the limit of active knee extension and flexion. In addition, the patient completed the Knee Society Knee and Function scores, Oxford Knee Score (OKS; 48 best, 0 worst), Knee injury and Osteoarthritis Outcome Score (KOOS JR; 100 best, 0 worst). Finally, they completed the Forgotten Joint Score (FJS; 100 best, 0 worst) at six weeks.

Discussion:

Comment 12: It would benefit the reader to further understand the implications the limitations of this study may have on the interpretation of the findings. Please expand this section accordingly. 

Authors’ response: As suggested we expanded the limitation section as follows:

The present study has several limitations that could affect the generalization of the results. First, the trochlear groove orientation and design of the MAD-FC reported in the present study might not apply to other femoral component designs. However, as presented in Figure 1, most of the MAD-FC from the major companies have a similar 6° trochlear groove orientation relative to vertical as the MAD-FC in the present study. Second, clinical follow-up at 6-weeks was short. But the aim of the study was to assess immediate potential adverse outcomes. Third, three AP scanograms were not measured because of asymmetric rotational projection of the two limbs. Obtaining optimal images required ongoing quality-control review. Fourth, because the study evaluated an anatomically shaped patella the patellofemoral effects of a domed patella implant and non-resurfaced patella remain unknown. Fifth, the PTA measured at six weeks is a non-dynamic static assessment of the patellofemoral joint at one flexion angle that is not a comprehensive assessment of patella kinematics. In addition, the difference in the time to obtain the skyline radiographic view from surgery being six weeks for the KAD-FC and a mean of 10 months for the MAD-FC could confound the interpretation. Finally, the differences in trochlear groove orientation and lateral coverage of the anterior femoral resection between component designs can not be explained by differences in post-operative coronal and axial alignment between limbs. The accuracy of setting the femoral component in varus-valgus, proximal-distal, internal-external, and anterior-posterior in each limb were within ± 0.5 mm due to the highly accurate caliper-verified KA with manual instruments (Table 4) [4,20]. In addition, different alignment philosophies that change the patient’s pre-arthritic joint lines and less accurate instrumentation could result in different orientation and coverage relative to KA.

Comment 13: Please also consider the possibility that the kinematics of the patella may be different at 6-weeks in each component, given the radiological differences. Assessing range of motion should not be a substitute to a biomechanical assessment of the patella's kinematics/kinetics. 

Authors’ response: As suggested we considered the possibility that the kinematics of the patella may be different at 6-weeks in each component. This has been added in the limitation section as follows:

Fifth, the PTA measured at six weeks is a non-dynamic static assessment of the patellofemoral joint at one flexion angle that is not a comprehensive assessment of patella kinematics. In addition, the difference in the time to obtain the skyline radiographic view from surgery being six weeks for the KAD-FC and a mean of 10 months for the MAD-FC could confound the interpretation.

Comment 14: Please rewrite the conclusions as it is not easy to follow. You should also explain what conclusions were drawn from your study, as opposed to merely stating that more research is necessary. Importantly, what impact might this particular study have? 

Authors’ response: As suggested we modified the conclusion as follows:

The KA TKA with a KAD-FC resulted in a trochlear groove lateral to the QLF that included the Q-angle in all patients, and negligible lateral undercoverage of the anterior femoral resection. These new radiographic parameters could be helpful when investigating femoral components designed for KA with the intent of promoting patellofemoral kinematics.

Tables and Figures:

Comment 15: In the table legends for Table 1 and 2 you can remove ‘Lists shows/lists the’ as these are not necessary. You should also ensure that all legends are descriptions of the content and not a discussion of the findings.

Authors’ response: As suggested the changes were done as follows:

‘Lists shows/lists the’ were removed.

All figure legends were changed to be description of the content when necessary.

Comment 16: Please explain why there is a p Value for the Complications in Table 3. Surely any complications would have been textual data rather than numeric?

Authors’ response: As suggested we removed the line on complications in Table 3, and we added the following clarification in the Results

No TKAs with the KAD- or MAD-FC had a complication at six weeks. Furthermore, the mean values for each post-operative clinical characteristic were not significantly different between the KAD- and MAD-FC (Table 3).

Reviewer 2 Report

Dear Authors,

Your article seems very interesting and well structured.

Some aspects have to be addressed. In particular, the discussion section must be improved. Your results are certainly clear, but You should better introduce the knee biomechanical characteristics and the clinical and rehabilitative lapels of Your findings. To do that, I suggest a brief integration usign the following references:

- Notarnicola A, Maccagnano G, Farì G, Bianchi FP, Moretti L, Covelli I, Ribatti P, Mennuni C, Tafuri S, Pesce V, Moretti B. Extracorporeal shockwave therapy for plantar fasciitis and gastrocnemius muscle: effectiveness of a combined treatment. J Biol Regul Homeost Agents. 2020 Jan-Feb;34(1):285-290. doi: 10.23812/19-347-L. PMID: 32191019.

Best regards and good luck

Author Response

Author’s response to reviewer #2

Dear Authors,

Your article seems very interesting and well structured.

Some aspects have to be addressed. In particular, the discussion section must be improved. Your results are certainly clear, but You should better introduce the knee biomechanical characteristics and the clinical and rehabilitative lapels of Your findings. To do that, I suggest a brief integration using the following references:

- Notarnicola A, Maccagnano G, Farì G, Bianchi FP, Moretti L, Covelli I, Ribatti P, Mennuni C, Tafuri S, Pesce V, Moretti B. Extracorporeal shockwave therapy for plantar fasciitis and gastrocnemius muscle: effectiveness of a combined treatment. J Biol Regul Homeost Agents. 2020 Jan-Feb;34(1):285-290. doi: 10.23812/19-347-L. PMID: 32191019.

Comment 1: Some aspects have to be addressed. In particular, the discussion section must be improved.

Authors’ response: As suggested we improved the several parts of the Discussion section as follows:

“The present study analyzed sequential bilateral unrestricted caliper-verified KA TKA performed with manual instruments on thirty-five patients with a KAD- and MAD-FC in opposite knees. The most important findings that the orientation of the trochlear groove of the KAD-FC was lateral to the QLF in all patients, thereby including the Q-angle and that the lateral undercoverage of the anterior femoral resection was negligible. In addition, monitoring the patellar tilt, complication rate, knee extension and flexion, and clinical outcome scores of the KAD-FC at six weeks did not detect any adverse or beneficial consequences.”

“Finally, monitoring the complication rate, knee extension, knee flexion, and clinical outcome scores at just six weeks was to detect any adverse consequences from using the KAD-FC, which were negligible. Unfortunately, the short follow-up and carry-over effect from implanting the KAD-FC after the MAD-FC in 34 of 35 patients prevents any conclusion concerning the effectiveness of the KAD-FC. However, the carry-over effect could be inconsequential as the preoperative to 6-week change was comparable for the KAD-FC vs. MAD-FC in terms of extension of 5° vs. 4°, flexion of -1° vs. -2°, Knee Society Knee Score 40 vs. 44 points, Knee Society Function Score of 9 vs. 8 points, and OKS 10 vs. 14 points (i.e., computed from Table 1 and 3). Furthermore, the difference between these pairs of motion and outcome scores was below the minimal clinically important difference.”

“This study has several limitations that could affect the generalization of the results. First, the trochlear groove orientation and design of the MAD-FC reported in the present study might not apply to other femoral component designs. However, as presented in Figure 1, most of the MAD-FC from the major companies have the same trochlear groove orientation as the MAD-FC studied. Further clinical studies involving other prostheses from different manufacturers are necessary to confirm these results. Second, clinical follow-up at 6-weeks was short. But the aim of the study was to assess immediate potential adverse outcomes. Longer follow-up is mandatory. Also, the results of the second surgery may potentially be biased by the outcomes of the first surgery. Third, three AP scanograms were not measured because of asymmetric rotational projection of the two limbs, which decreased the number of patients with the assessment of the trochlear groove orientation relative to the QLF and the patient’s Q-angle. Fourth, patella shift was not measured because the design of the trochlea is different between the two components. Thus, the measurement technique is not applicable. Fifth, with the difference  in the observed results, kinematics of the patella may be different and specific biomechanical assessments of the patella’s kinematics are necessary. Finally, the results are not explained by differences in post-operative alignment between the paired limbs as the hip-knee-ankle angle and distal lateral femoral angle were comparable, which is due to the highly accurate caliper-verified KA with manual instruments (Table 4) [4,19].”

Comment 2: Your results are certainly clear, but You should better introduce the knee biomechanical characteristics and the clinical and rehabilitative lapels of Your findings

Authors’ response: As suggested we introduced the knee biomechanics of the tochlear groove – quadriceps line of force as they relate to patellofemoral tracking and the Q-angle by modifying the test and changing Figure 3 as follows:

When the trochlear groove is lateral or medial relative to the QLF, the angle is denoted + or –, and the femoral component includes or excludes the patient’s Q-angle, respectively. Therefore one strategy to promote inclusion of the Q-angle is to design a KAD-FC with a wide lateral trochlear orientation that lies lateral to the QLF across the wide range of femoral and tibial phenotypes (Figure 3)[15].

Figure 3: Schematics show the 6° trochlear groove (pink line) of the MAD-FC (left) and the 20.5° trochlear groove of the KAD-FC (right) evaluated in opposite knees in each patient enrolled in the present study. The TG-QLF angle on the scanogram was -3° for the MAD-FC (left) and 17° for the KAD-FC (right).

:

 Comment 3: I suggest a brief integration using the following references:

- Notarnicola A, Maccagnano G, Farì G, Bianchi FP, Moretti L, Covelli I, Ribatti P, Mennuni C, Tafuri S, Pesce V, Moretti B. Extracorporeal shockwave therapy for plantar fasciitis and gastrocnemius muscle: effectiveness of a combined treatment. J Biol Regul Homeost Agents. 2020 Jan-Feb;34(1):285-290. doi: 10.23812/19-347-L. PMID: 32191019.

Authors’ response: We enjoyed reading the above well-written paper. However, the paper discusses treating plantar fasciitis and uses shock-wave therapy of the plantar fascia and the gastrocnemius to do so. Unfortunately, we were unable to find a way to add this reference in our study of TKA, as our patients did not have plantar fasciitis.
